# Tsallis Distribution as a Λ-Deformation of the Maxwell–Jüttner Distribution

**DOI:** 10.3390/e26030273

**Published:** 2024-03-21

**Authors:** Jean-Pierre Gazeau

**Affiliations:** Centre National de la Recherche Scientifique (CNRS), Astroparticule et Cosmologie, Université Paris Cité, F-75013 Paris, France; gazeau@apc.in2p3.fr

**Keywords:** Maxwell–Jüttner distribution, Tsallis distribution, de Sitter quantum field, ΛCDM standard model

## Abstract

Currently, there is no widely accepted consensus regarding a consistent thermodynamic framework within the special relativity paradigm. However, by postulating that the inverse temperature 4-vector, denoted as β, is future-directed and time-like, intriguing insights emerge. Specifically, it is demonstrated that the *q*-dependent Tsallis distribution can be conceptualized as a de Sitterian deformation of the relativistic Maxwell–Jüttner distribution. In this context, the curvature of the de Sitter space-time is characterized by $\sqrt{\Lambda /3}$, where Λ represents the cosmological constant within the ΛCDM standard model for cosmology. For a simple gas composed of particles with proper mass *m*, and within the framework of quantum statistical de Sitterian considerations, the Tsallis parameter *q* exhibits a dependence on the cosmological constant given by $q=1+\ell_{c}\sqrt{\Lambda }/n$, where $\ell_{c}=\hbar /mc$ is the Compton length of the particle and n is a positive numerical factor, the determination of which awaits observational confirmation. This formulation establishes a novel connection between the Tsallis distribution, quantum statistics, and the cosmological constant, shedding light on the intricate interplay between relativistic thermodynamics and fundamental cosmological parameters.

## 1. Preamble: Temperature, Heat, and Entropy, That Obscure Objects of Desire

It is opportune to start out this contribution by quoting what de Broglie wrote in Ref. [1] about the relation between entropy invariance and relativistic variance of temperature (translated from French):


*It is well known that entropy, alongside the space-time interval, electric charge, and mechanical action, is one of the fundamental “invariants” of the theory of relativity. To convince oneself of this, it is enough to recall that, according to Boltzmann, the entropy of a macroscopic state is proportional to the logarithm of the number of microstates that realize that state. To strengthen this reasoning, one can argue that, on the one hand, the definition of entropy involves a integer number of microstates, and, on the other hand, the transformation of entropy during a Galilean reference frame change must be expressed as a continuous function of the relative velocity of the reference frames. Consequently, this continuous function is necessarily constant and equal to unity, which means that entropy is constant.*


Let us now give more insights about what “relativistic thermodynamics” could be. In relativistic thermodynamics (i.e., in accordance with special relativity), there exist three points of view [2], distinguished from the way heat ΔQ and temperature *T* transform under a Lorentz boost from frame $R_{0}$ (e.g., laboratory) to comoving frame R with velocity $v=v\hat{n}$ relative to $R_{0}$ and Lorentz factor
(1)$$\gamma \left(v\right)=\frac{1}{\sqrt{1-v^{2}/c^{2}}}.$$

(**a**)The covariant viewpoint (Einstein [3], Planck [4], de Broglie [1] …),
(2)$$\Delta Q=\Delta Q_{0}\gamma^{-1}\;,\;T=T_{0}\gamma^{-1}.$$(**b**)The anti-covariant one (Ott [5], Arzelies [6], …),
(3)$$\Delta Q=\Delta Q_{0}\gamma \;,\;T=T_{0}\gamma .$$(**c**)The invariant one, “nothing changes” (Landsberg [7,8], …),
(4)$$\Delta Q=\Delta Q_{0}\;,\;T=T_{0}.$$

Also note that, for some authors (Landsberg [9], Sewell [10], …), “there is no meaningful law of temperature under boosts”.

Nevertheless, more recent approaches (e.g., Ref. [11]) show that there is a covariant relativistic thermodynamics with proper absolute temperature in full agreement with relativistic hydrodynamics.

In this paper, we adopt the viewpoint in Section 1 and review de Broglie’s arguments in Section 2. In Section 3, we remind you of the construction of the so-called Maxwell–Jüttner distribution presented by Synge in Ref. [12]. In Section 4, we then present the de Sitter space-time, its geometric description as a hyperboloid embedded in the 1+4 Minkowski space-time, and give some insights of the fully covariant quantum field theory of free scalar massive elementary systems propagating on this manifold. In Section 5, we then develop our arguments in favor of a novel connection between the Tsallis distribution, quantum statistics, and the cosmological constant, shedding light on the intricate interplay between relativistic thermodynamics and fundamental cosmological parameters. A few comments end our paper in Section 6.

## 2. Relativistic Covariance of Temperature According to de Broglie (1948)

Here, we give an account of the de Broglie arguments given in Ref. [1] in favor of the covariant viewpoint (a).

Let us consider a body B with proper frame $R_{0}$, and total proper mass $M_{0}$. It is assumed to be in thermodynamical equilibrium with temperature $T_{0}$ and fixed volume $V_{0}$ (e.g., a gas enclosed with surrounding rigid wall). Let us then observe B from an inertial frame R, in which B has constant velocity $v=v\hat{n}$ relative to $R_{0}$. We suppose that a source in R provides B with heat ΔQ. In order to keep the velocity v of B constant, work *W* has to be performed on B. Its proper mass is consequently modified $M_{0}\to M_{0}^{\prime }$. Then, from energy conservation,
(5)$$\left(M_{0}^{\prime }-M_{0}\right)\gamma c^{2}=\Delta Q+W\;,\;\gamma =\gamma \left(v\right)=\frac{1}{\sqrt{1-v^{2}/c^{2}}}\;,$$
and the relativistic second Newton law,
(6)$$\Delta P=M_{0}^{\prime }\gamma v-M_{0}\gamma v=\int Fdt=\frac{1}{v}\int Fvdt=\frac{W}{v}\;,$$
we derive
(7)$$\Delta Q=\frac{c^{2}}{v^{2}}\gamma^{-2}W=\left(M_{0}^{\prime }-M_{0}\right)c^{2}\gamma^{-2}\;.$$

In frame $R_{0}$, there is no work performed (the volume is constant), there is just transmitted heat $\Delta Q_{0}=\left(M_{0}^{\prime }-M_{0}\right)c^{2}$. By comparison with (Equation 7), one infers that heat transforms as
(8)$$\Delta Q=\Delta Q_{0}\gamma^{-1}\;.$$

Since the entropy $S=\int \frac{dQ}{T}$ is relativistic invariant, $S=S_{0}$, temperature finally transforms as
(9)$$T=T_{0}\gamma^{-1}$$

## 3. Maxwell–Jüttner Distribution

We now present a relativistic version of the Maxwell–Boltzmann distribution for simple gases, namely the Maxwell–Jüttner distribution [13,14,15]. We follow the derivation given by Synge in Ref. [12]; see also Ref. [16], and the recent article [17] for a comprehensive list of references. Note that this distribution is defined on the mass hyperboloid, and not expressed in terms of velocities (see the recent [18] and references therein).

Our notations [19] for event four-vector $\underline{x}$ in the Minkowskian space-time $M_{1,3}$ and for four-momentum $\underline{k}$ are the following:(10)$$M_{1,3}∋\underline{x}=\left(x^{\mu }\right)=\left(x^{0}=x_{0},x^{i}=-x_{i},\;i=1,2,3\right)\equiv \left(x^{0},x\right)\;,$$
equipped with the metric $ds^{2}={\left(dx^{0}\right)}^{2}-dx\cdot x\equiv g_{\mu \nu }dx^{\mu }dx^{\nu }$, $g_{\mu \nu }=\mathrm{diag}\left(1,-1,-1,-1\right)$,
(11)$$\underline{k}=\left(k^{\mu }\right)=\left(k^{0},k\right).$$

The Minkowskian inner product is noted by:(12)$$\;\underline{x}\cdot \underline{x^{\prime }}=g_{\mu \nu }x^{\mu }{x^{\prime }}^{\nu }=x^{\mu }x_{\mu }^{\prime }=x^{0}{x^{\prime }}^{0}-x\cdot x^{\prime }.$$

Let $\underline{k}$ be four-momentum, pointing toward point *A* of the mass shell hyperboloid $V_{m}^{+}=\left\{\underline{k}\;,\;\underline{k}\cdot \underline{k}=m^{2}c^{2}\right\}$, and an infinitesimal hyperbolic interval at *A*, with length
(13)dσ=mcdω,
where $d\omega =\frac{d^{3}k}{k_{0}}$ is the Lorentz-invariant element on $V_{m}^{+}$. Given a time-like unit vector $\underline{n}$, and a straight line Δ passing through the origin and orthogonal (in the $M_{1,3}$ metric sense) to $\underline{n}$, denote by dΩ the length of the projection of dσ on Δ along $\underline{n}$. As is illustrated in Figure 1, one easily proves that
(14)$$d\Omega =|\underline{k}\cdot \underline{n}|\;d\omega \;\left(=d^{3}k\;\mathrm{if}\;\underline{n}=\left(1,0\right)\right).$$

The sample population consists of those particles with world lines cutting the infinitesimal space-like segment dΣ orthogonal to the time-like unit vector $\underline{n}$, as is shown in Figure 2.

Every particle that traverses the segment C of the null cone between *M* and dΣ must also traverse dΣ (causal cone). Consequently, regardless of the collisions that take place within the infinitesimal region R bounded by *M*, the segment of the light cone C, and dΣ, the number of particles crossing Σ, is predetermined as the number crossing C:(15)$$\nu =\underline{N}\cdot \underline{n}\;d\Sigma =d\Sigma \;\int_{V_{m}^{+}}N\left(\underline{x},\underline{k}\right)\;d\Omega \;,$$
where $\underline{N}$ is the numerical-flux four-vector and $N(\underline{x},\underline{k})$ is the distribution function. By the conservation of four-momentum at each collision in a simple gas, the flux of four-momentum across dΣ is predetermined as the flux across C,
(16)$$T_{\mu }\cdot \underline{n}\;d\Sigma =d\Sigma \;\int_{V_{m}^{+}}N\left(\underline{x},\underline{k}\right)\;ck_{\mu }d\Omega \;,$$
where $\underline{\underline{T}}=\left(T_{\mu \nu }\right)$ is the energy-momentum tensor.

The most probable distribution function N at *M* is that which maximizes the following entropy integral:(17)$$F=-d\Sigma \;\int_{V_{m}^{+}}N\left(\underline{x},\underline{k}\right)\;\log N\left(\underline{x},\underline{k}\right)\;d\Omega .$$

Variational calculus with five Lagrange $\underline{x}$-dependent multipliers α and $\eta_{\mu }$ associated with constraints on ν and $T_{\mu }\cdot \underline{n}$, respectively, leads to the solution
(18)$$N\left(\underline{x},\underline{k}\right)=C\left(\underline{x}\right)\;\exp\left(-\underline{\eta }\left(\underline{x}\right)\cdot \underline{k}\right)\;,\;C=e^{\alpha -1}.$$

Scalar *C* and time-like four-vector $\underline{\eta }$ are determined by the constraints on $\nu =\underline{N}\cdot \underline{n}\;d\Sigma$ and $T_{\mu }\cdot \underline{n}\;d\Sigma$:(19)$$C\int_{V_{m}^{+}}k_{\mu }\;e^{-\underline{\eta }\cdot \underline{k}}\;d\omega =N_{\mu }\;,\;C\int_{V_{m}^{+}}ck_{\mu }k_{\nu }\;e^{-\underline{\eta }\cdot \underline{k}}\;d\omega =T_{\mu \nu }.$$
established by taking into account that $\underline{n}$ is arbitrary.

With the equations of conservation
(20)$$\underline{\partial }\cdot \underline{N}=0\;,\;\underline{\partial }\cdot T_{\mu }=0\;,$$

We finally obtain as many equations as the 19 functions of $\underline{x}$: $C,\underline{\eta },\underline{N},T$. The following partition function is essential for all relevant calculations.
(21)$$Z\left(\eta \right):=\int_{V_{m}^{+}}e^{-\underline{\eta }\cdot \underline{k}}\;\frac{d^{3}k}{k_{0}}=\frac{4\pi mc}{\sqrt{\underline{\eta }\cdot \underline{\eta }}}K_{1}\left(mc\sqrt{\underline{\eta }\cdot \underline{\eta }}\right)$$
where $K_{\nu }$ is the modified Bessel function [20]. Hence, the components of the numerical flux four-vector $\underline{N}$ and of the energy tensor $\underline{\underline{T}}$ in (Equation 19) are given in terms of derivatives of *Z* and, finally, in terms of Bessel functions by
(22)$$\begin{array}{cc} N_{\mu } & =-C\frac{\partial Z}{\partial \eta^{\mu }}=C\frac{4\pi m^{2}c^{2}\eta_{\mu }}{\underline{\eta }\cdot \underline{\eta }}K_{2}\left(mc\sqrt{\underline{\eta }\cdot \underline{\eta }}\right), \end{array}$$
(23)$$\begin{array}{cc} T_{\mu \nu } & =Cc\frac{\partial^{2}Z}{\partial \eta^{\mu }\partial \eta^{\nu }}=C4\pi m^{2}c^{3}\left[mc\frac{K_{3}\left(mc\sqrt{\underline{\eta }\cdot \underline{\eta }}\right)}{{\left(\underline{\eta }\cdot \underline{\eta }\right)}^{3/2}}\eta_{\mu }\eta_{\nu }-\frac{K_{2}\left(mc\sqrt{\underline{\eta }\cdot \underline{\eta }}\right)}{\underline{\eta }\cdot \underline{\eta }}g_{\mu \nu }\right]. \end{array}$$

For a simple gas consisting of material particles of proper mass *m*, the components of the energy–momentum tensor $\underline{\underline{T}}$ are given by
(24)$$T_{\mu \nu }=\left(\rho +p\right)u_{\mu }u_{\nu }-pg_{\mu \nu }\;,$$
where ρ is the mean density, *p* is the pressure, and $\underline{u}=\left(u_{\mu }=\frac{dx_{\mu }}{ds}\right)$, $\underline{u}\cdot \underline{u}=1$, is the mean four-velocity of the fluid. Hence, by identification with (23), Synge [12] proved that *a relativistic gas consisting of material particles of proper mass m is a perfect fluid* through the relations: (25)$$\begin{array}{cc} u_{\mu } & =\frac{\eta_{\mu }}{\sqrt{\underline{\eta }\cdot \underline{\eta }}}\;, \end{array}$$(26)$$\begin{array}{cc} \rho +p & =C4\pi m^{3}c^{4}\frac{K_{3}\left(mc\sqrt{\underline{\eta }\cdot \underline{\eta }}\right)}{\sqrt{\underline{\eta }\cdot \underline{\eta }}}\;, \end{array}$$(27)$$\begin{array}{cc} p & =C4\pi m^{2}c^{3}\frac{K_{2}\left(mc\sqrt{\underline{\eta }\cdot \underline{\eta }}\right)}{\underline{\eta }\cdot \underline{\eta }}\;. \end{array}$$

From (26) and (27), we derive the expression of the density:(28)$$\rho =C\frac{4\pi m^{3}c^{4}}{\sqrt{\underline{\eta }\cdot \underline{\eta }}}\frac{K_{1}\left(mc\sqrt{\underline{\eta }\cdot \underline{\eta }}\right)+K_{3}\left(mc\sqrt{\underline{\eta }\cdot \underline{\eta }}\right)}{2}=-C\frac{4\pi m^{3}c^{4}}{\sqrt{\underline{\eta }\cdot \underline{\eta }}}K_{2}^{\prime }\left(mc\sqrt{\underline{\eta }\cdot \underline{\eta }}\right).$$

Let us define the invariant quantity, i.e., the projection of the numerical flux (57) along the four-velocity of the fluid,
(29)$$N_{0}=\underline{N}\cdot \underline{u}=C\frac{4\pi m^{2}c^{2}}{\sqrt{\underline{\eta }\cdot \underline{\eta }}}K_{2}\left(mc\sqrt{\underline{\eta }\cdot \underline{\eta }}\right)\;.$$

This expression, which represents the number of particles per unit length (“numerical density”) in the rest frame of the fluid ($u_{0}=1$), allows us to determine the function $C=C(\underline{x})$, and to eventually write Distribution (Equation 18) as:(30)$$N\left(\underline{x},\underline{k}\right)=\frac{N_{0}}{m^{2}ck_{B}T_{a}K_{2}\left(mc^{2}/k_{B}T_{a}\right)}\exp\left(-\frac{c\underline{u}\cdot \underline{k}}{k_{B}T_{a}}\right).$$

The term $T_{a}:=c/\left(k_{B}\sqrt{\underline{\eta }\cdot \underline{\eta }}\right)$, where $k_{B}$ is the Boltzmann constant, is a “relativistic” absolute temperature. It is precisely the relativistic invariant, which might fit pointview (**c**).

Note that, with this expression, (27) reads as the usual gas law:(31)$$p=N_{0}k_{B}T_{a}.$$

The Maxwell–Boltzmann non relativistic distribution (in the space of momenta) is recovered by considering the limit at $k_{B}T_{a}\ll mc^{2}$ in the rest frame of the fluid:(32)$$\begin{array}{c} K_{2}\left(\frac{mc^{2}}{k_{B}T_{a}}\right)\approx \sqrt{\frac{\pi k_{B}T_{a}}{2mc^{2}}}e^{-\frac{mc^{2}}{k_{B}T_{a}}}\\ \Rightarrow N(\underline{x},\underline{k})\\ \approx N_{0}{\left(2\pi mk_{B}T_{a}\right)}^{-3/2}\exp\left(-\frac{k_{0}c-mc^{2}}{k_{B}T_{a}}\right)\approx N_{0}{\left(2\pi mk_{B}T_{a}\right)}^{-3/2}\exp\left(-\frac{k^{2}}{2mk_{B}T_{a}}\right). \end{array}$$

### Inverse Temperature Four-Vector

The found distribution (Equation 30) on the Minkowskian mass shell for a simple gas consisting of particles of proper mass *m* leads us to introduce the relativistic thermodynamic, future directed, time-like four-coldness vector $\underline{\beta }$, as the four-version of the reciprocal of the thermodynamic temperature (see also Ref. [2]):(33)$$\frac{c\underline{u}}{k_{B}T_{a}}\equiv \underline{\beta }=\left(\beta^{0}=\beta_{0}>0,\beta^{i}=-\beta_{i}\right)=\left(\beta_{0},\beta \right),$$
with *absolute coldness* as relativistic invariant,
(34)$$\sqrt{\underline{\beta }\cdot \underline{\beta }}=\frac{c}{k_{B}T_{a}}\equiv \beta_{a}\;.$$

It is precisely the way the component $\beta_{0}$ transforms under a Lorentz boost, $\beta_{0}^{\prime }=\gamma \left(v\right)\left(\beta_{0}-v\cdot \beta /c\right)$, which explains the way the temperature transforms à la de Broglie, $T\mapsto T^{\prime }=T\gamma^{-1}$. So, in the follow-up, we call Maxwell–Jüttner distribution the following relativistic invariant:(35)$$N\left(\underline{\beta },\underline{k}\right)=\frac{N_{0}}{mcK_{1}\left(mc\beta_{a}\right)}\exp\left(-\underline{\beta }\cdot \underline{k}\right),$$
where the space-time dependence holds through the coldness four-vector coldness field $\underline{\beta }=\underline{\beta }\left(\underline{x}\right)$.

## 4. de Sitter Material

We now turn our attention to the de Sitter (dS) space-time and some important features of a dS covariant quantum field theory.

### 4.1. de Sitter Geometry

The de Sitter space-time can be viewed as a hyperboloid embedded in a five-dimensional Minkowski space $M_{1,4}$ with metric $g_{\alpha \beta }=$diag(1,−1,−1,−1,−1) (see Figure 3). Of course, one should keep in mind that all choices of one point in the manifold as an origin are physically equivalent, as are the points of the Minkowski space-time $M_{1,3}$.
(36)$$M_{R}\equiv \left\{x\in R^{5};\;x^{2}=g_{\alpha \beta }\;x^{\alpha }x^{\beta }=-R^{2}\right\},\;\alpha ,\beta =0,1,2,3,4\;,$$
where the pseudo-radius *R* (or inverse of curvature) is given by $R=\sqrt{\frac{3}{\Lambda }}$ within the cosmological ΛCDM standard model. The de Sitter symmetry group is the group SO_0_(1,4) of proper (i.e., det.=1) and orthochronous (to be precised later) transformations of the manifold (Equation 36). This group has ten (Killing) generators $K_{\alpha \beta }=x_{\alpha }\partial_{\beta }-x_{\beta }\partial_{\alpha }$.

### 4.2. Flat Minkowskian Limit of de Sitter Geometry

Let us choose the global coordinates $ct\in R,n\in S^{2},r/R\in \left[0,\pi \right]$ for the dS manifold $M_{R}$. They are defined by:(37)$$\begin{array}{cc} M_{R}∋x & =\left(x^{0},x^{1},x^{2},x^{3},x^{4}\right)\equiv \left(x^{0},x,x^{4}\right)\\ \; & =\left(R\sinh(ct/R),\;R\cosh(ct/R)\sin(r/R)n,R\cosh(ct/R)\cos(r/R)\right)\equiv x\left(t,x\right). \end{array}$$

At the limit R→∞, and the manifold $M_{R}\to M_{1,3}$, the Minkowski space-time tangent to $M_{R}$ at, say, the de Sitter point $O_{dS}=\left(0,0,R\right)$, chosen as the origin, since
(38)$$M_{R}∋x\underset{R\to \infty }{\approx }\left(ct,r=r\;n,R\right)\equiv \left(\underline{\ell },R\right)\;,\;\underline{\ell }\in M_{1,3}.$$

At this limit, the de Sitter group becomes the Poincaré group:(39)$$\lim_{R\to \infty }{\mathrm{SO}}_{0}\left(1,4\right)=P_{+}^{↑}\left(1,3\right)=M_{1,3}⋊{\mathrm{SO}}_{0}\left(1,3\right).$$

Consistently, the ten de Sitter Killing generators contract (in the Wigner–Inönü sense) to their Poincaré counterparts $K_{\mu \nu }$, $\Pi_{\mu }$, μ=0,1,2,3, after rescaling the four $K_{4\mu }⟶\Pi_{\mu }=K_{4\mu }/R$.

### 4.3. de Sitter Plane Waves as Binomial Deformations of Minkowskian Plane Waves

The de Sitter (scalar) plane waves are defined in [21] as
(40)$$\phi_{\tau ,\xi }\left(x\right)={\left(\frac{x\cdot \xi }{R}\right)}^{\tau }\;,\;x\in M_{R}\;,\;\xi \in C_{1,4}\;,$$
where $C_{1,4}=\left\{\xi \in R^{5}\;,\;\xi \cdot \xi =0\right\}$ is the null cone in $M_{1,4}$. They are solutions of the Klein–Gordon-like equation
$$\frac{1}{2}M_{\alpha \beta }M^{\alpha \beta }\phi_{\tau ,\xi }\left(x\right)\equiv R^{2}{□}_{R}\phi_{\tau ,\xi }\left(x\right)=\tau \left(\tau +3\right)\phi_{\tau ,\xi }\left(x\right)\;,$$
where $M_{\alpha \beta }=-i\left(x_{\alpha }\partial_{\beta }-x_{\beta }\partial_{\alpha }\right)$ is the quantum representation of the Killing vector $K_{\alpha \beta }$, and ${□}_{R}$ stands for the d’Alembertian operator on $M_{R}$. For the values
(41)$$\tau =-\frac{3}{2}+i\nu \;,\;\nu \in R\;,$$
they describe free quantum motions of “massive” scalar particles on $M_{R}$. The term “massive” is justified by the flat Minkowskian limit R→∞, i.e., Λ→0. This limit is understood as follows.

(**i**)First, one has the Garidi [22] relation between proper mass *m* (curvature independent) of the spinless particle and the parameter ν≥0:
(42)$$m=\frac{\hbar }{Rc}{\left[\nu^{2}+\frac{1}{4}\right]}^{1/2}\Leftrightarrow \nu =\sqrt{\frac{R^{2}m^{2}c^{2}}{\hbar^{2}}-\frac{1}{4}}\underset{R\;\mathrm{large}}{\approx }\frac{Rmc}{\hbar }=\frac{mc}{\hbar }\sqrt{\frac{3}{\Lambda }}.$$The quantity $\frac{\hbar c\nu }{R}$ is a kind of *at rest de Sitterian energy*, which is distinct of the proper mass energy $mc^{2}$ if Λ≠0.(**ii**)Then, with the mass shell parameterization $\xi =\left(\xi^{0}=\frac{k_{0}}{mc},\xi =\frac{k}{mc},\xi^{4}=1\right)\in C_{1,4}^{+}$, one obtains at the limit R→∞:
(43)$$\phi_{\tau ,\xi }\left(x\right)={\left(x\cdot \xi /R\right)}^{-3/2+i\nu }\underset{R\to \infty }{\to }e^{i\underline{k}\cdot \underline{\ell }/\hbar }\;,\;\underline{\ell }=\left(ct,r\right).$$

This relation allows us to consider Function (Equation 40) as deformation of the plane waves propagating in the Minkowskian space-time $M_{1,4}$. This pivotal property justifies the name “dS plane waves” granted to Function (Equation 40).

### 4.4. Analytic Extension of dS Plane Waves for dS QFT

The dS plane waves $\phi_{\tau ,\xi }\left(x\right)={\left(\frac{x\cdot \xi }{R}\right)}^{\tau }$, τ=−3/2+iν, are not defined on all $M_{R}$, due to the possible change of sign of x·ξ. A solution to this drawback is found through the extension to the tubular domains in the complexified hyperboloid $M_{R}^{C}=\left\{z=x+iy\in C^{5}\;,\;z^{2}=g_{\alpha \beta }\;z^{\alpha }z^{\beta }=-R^{2}\right.$ or, equivalently, $\left.x^{2}-y^{2}=-R^{2}\;,\;x\cdot y=0\right\}$:(44)$$T^{\pm }:=T^{\pm }\cap M_{R}^{C}\;,\;T^{\pm }:=M_{1,4}+iV^{\pm },$$
where the forward and backward light cones $V^{\pm }:=\left\{x\in M_{1,4}\;,\;x^{0}≷\sqrt{x^{2}+{\left(x^{4}\right)}^{2}}\right\}$ allow for a causal ordering in $M_{1,4}$.

Then, the extended plane waves $\phi_{\tau ,\xi }\left(z\right)={\left(\frac{z\cdot \xi }{R}\right)}^{\tau }$ are globally defined for $z\in T^{\pm }$ and $\xi \in C_{1,4}^{+}$.

These analytic extensions allow for a consistent QFT for free scalar fields on $M_{R}$: the two-point Wightman function $W_{\nu }\left(x,x^{\prime }\right)=\langle \Omega ,\phi \left(x\right)\phi \left(x^{\prime }\right)\Omega \rangle$ can be extended to the complex covariant, maximally analytic, two-point function having the spectral representation in terms of these extended plane waves:(45)$$W_{\nu }\left(z,z^{\prime }\right)=c_{\nu }\int_{V_{m}^{+}\cup V_{m}^{-}}{\left(z\cdot \xi \right)}^{-3/2+i\nu }{\left(\xi \cdot z^{\prime }\right)}^{-3/2-i\nu }\;\frac{dk}{k_{0}}\;,\;z\in T^{-}\;,\;z^{\prime }\in T^{+}.$$

Details are found in Ref. [21] and in the recent volume [23].

### 4.5. KMS Interpretation of $W_{\nu }\left(z,z^{\prime }\right)$ Analyticity

From the analyticity of $W_{\nu }\left(z,z^{\prime }\right)$, we deduce that $W_{\nu }\left(x,x^{\prime }\right)$ defines a 2iπR/c periodic analytic function of *t*, whose domain is the periodic cut plane
(46)Cx,x′cut={t∈C,Im(t)≠2nπR/c,n∈Z}∪{t,t−2inπR/c∈Ix,x′,n∈Z},
where $I_{x,x^{\prime }}$ is the real interval on which ${\left(x-x^{\prime }\right)}^{2}<0$. Hence, $W_{\nu }\left(z,z^{\prime }\right)$ is analytic in the strip
(47){t∈C,0<Im(t)<2iπR/c},
and satisfies
(48)$$W_{\nu }\left(x^{\prime }\left(t+t^{\prime },x\right),x\right)=\lim_{\epsilon \to 0^{+}}W_{\nu }\left((x,x^{\prime }\left(t+t^{\prime }+2i\pi R/c-i\epsilon ,x\right)\right)\;,\;t^{\prime }\in R\;.$$

This is a KMS relation at (∼ Hawking) temperature
(49)$$T_{\Lambda }=\frac{\hbar c}{2\pi k_{B}R}:=\frac{\hbar c}{2\pi k_{B}}\sqrt{\frac{\Lambda }{3}}.$$

## 5. de Sitterian Tsallis Distribution

### 5.1. Tsallis Entropy and Distribution: A Short Reminder

Given a discrete (resp. continuous) set of probabilities $\left\{p_{i}\right\}$ (resp. continuous x↦p(x)) with $\sum_{i}p_{i}=1$ (resp. ∫p(x)dx=1), and a real *q*, the Tsallis entropy [24] is defined as
(50)$$S_{q}\left(p_{i}\right)=k\frac{1}{q-1}\left(1-\sum_{i}p_{i}^{q}\right)\;\mathrm{resp}.\;S_{q}\left[p\right]=\frac{1}{q-1}\left(1-\int {\left(p\left(x\right)\right)}^{q}dx\right).$$

As q→1, $S_{q}\left(p_{i}\right)\to S_{\mathrm{BG}}\left(p\right)=-k\sum_{i}p_{i}\ln p_{i}$ (Boltzmann–Gibbs). The Tsallis entropy is non additive for two independent systems, *A* and *B*, for which p(A∪B)=p(A)p(B), $S_{q}\left(A\cup B\right)=S_{q}\left(A\right)+S_{q}\left(B\right)+\left(1-q\right)S_{q}\left(A\right)S_{q}\left(B\right)$. A *Tsallis distribution* is a probability distribution derived from the maximization of the Tsallis entropy under appropriate constraints. The so-called *q*-exponential Tsallis distribution has the probability density function
(51)$$\left(2-q\right)\lambda {\left[1-\left(1-q\right)\lambda x\right]}^{1/(1-q)}\equiv \left(2-q\right)\lambda e_{q}\left(-\lambda x\right)\;,$$
where q<2 and λ>0 (rate) arise from the maximization of the Tsallis entropy under appropriate constraints, including constraining the domain to be positive. More details are given, for instance, in Ref. [25].

Let us now show how the Tsallis distribution can be viewed as a Λ-deformation of the Maxwell–Jüttner distribution.

### 5.2. Coldness in de Sitter

Analogous with the de Sitter plane waves, we introduce the following distributions on the subset $∼V_{m}^{+}$ of the null cone $C_{1,4}^{+}=\left\{\xi \in M_{1,4}\;,\;\xi \cdot \xi =0\;,\;\xi^{0}>0\right\}$:(52)$$\phi_{\tau ,\xi }\left(x\right)={\left(\frac{b\cdot \xi }{B}\right)}^{\tau }\;,\;b\in M_{B}\;,\;\xi =\left(\frac{k^{0}}{mc}>0,\frac{k}{mc},-1\right),$$
where one should note the negative value −1 for $\xi_{4}$, and
(53)$$M_{B}\equiv \left\{b\in M_{1,4}\;,\;b^{2}=g_{\alpha \beta }\;b^{\alpha }b^{\beta }=-B^{2}\right\},\;\;\alpha ,\beta =0,1,2,3,4\;,$$
is the manifold of the “de Sitterian five-vector coldness fields” b=b(x).

Like for $M_{R}$, we use global coordinates on $M_{B}$:(54)$$\beta^{0}\in R\;,\;\beta =∥\beta ∥n\in R^{3}\;,\;∥\beta ∥/B\in \left[0,\pi \right]\;,$$
with
(55)$$\begin{array}{c} M_{B}∋b\equiv b\left(\underline{\beta }\right)=\left(b^{0},b^{1},b^{2},b^{3},b^{4}\right)\equiv \left(b^{0},b,b^{4}\right)\\ =\left(B\sinh\left(\beta^{0}/B\right),\;B\cosh\left(\beta^{0}/B\right)\sin\left(∥\beta ∥/B\right)n,-B\cosh\left(\beta^{0}/B\right)\cos\left(∥\beta ∥/B\right)\right), \end{array}$$
in such a way that at large *B* we recover the Minkowskian coldness $\underline{\beta }$:$$M_{B}∋b\underset{B\to \infty }{∼}\left(\underline{\beta },B\right).$$

We now need to connect the de Sitterian coldness scale *B* with Λ. Inspired by the relativistic invariant $\beta_{a}=\frac{c}{k_{B}T_{a}}$ and the KMS temperature $T_{\Lambda }=\frac{\hbar c}{2\pi k_{B}}\sqrt{\frac{\Lambda }{3}}$, we write
(56)$$B\propto \frac{2\pi }{\hbar }\sqrt{\frac{3}{\Lambda }},\;i.e.,\;B=\frac{n}{\hbar \sqrt{\Lambda }}\;,$$
where n is a numerical factor. Note that, with the values
$$\Lambda_{\mathrm{current}}=1.1056\times 10^{-52}\;m^{-2}\;,\;\hbar =1.054571817\ldots \times 10^{-34}\;J\;s\;,$$
one obtains $B\approx 0.9\times 10^{60}\;n$ SI (inverse of a momentum).

### 5.3. A de Sitterian Tsallis Distribution

We now consider the distribution on $M_{B}\times V_{m}^{+}$ with $B=\frac{n}{\hbar \sqrt{\Lambda }}$:(57)$$N\left(b,\underline{k}\right)=C_{B}{\left(\frac{b\cdot \xi }{B}\right)}^{-mcB}=C_{B}{\left(\frac{b^{0}}{B}\frac{k^{0}}{mc}-\frac{b}{B}\cdot \frac{k}{mc}+\frac{b^{4}}{B}\right)}^{-mcB}.$$
$$b\in M_{B}\;,\;\xi =\left(\frac{k^{0}}{mc}>0,\frac{k}{mc},-1\right),$$
where the constant $C_{B}$ involves an associated Legendre function of the First Kind [26].

With the global coordinates (Equation 55), and with the constraint $\beta^{0}/B\in \left[0,\pi /2\right)$, the distribution $N(b,\underline{k})$ reads
(58)$$\begin{array}{c} N(b,\underline{k})\\ =C_{B}{\left(\cosh\left(\beta^{0}/B\right)\cos\left(∥\beta ∥/B\right)+\sinh\left(\beta^{0}/B\right)\;\frac{k^{0}}{mc}-\cosh\left(\beta^{0}/B\right)\sin\left(∥\beta ∥/B\right)\frac{n\cdot k}{mc}\right)}^{-mcB}\\ =C_{B}\exp\left[-mcB\log\left(\cosh\left(\beta^{0}/B\right)\cos\left(∥\beta ∥/B\right)\right)\right]\\ \times \exp\left[-mcB\log\left(1+\frac{\sinh\left(\beta^{0}/B\right)\;\frac{k^{0}}{mc}-\cosh\left(\beta^{0}/B\right)\sin\left(∥\beta ∥/B\right)\frac{n\cdot k}{mc}}{\cosh\left(\beta^{0}/B\right)\cos\left(∥\beta ∥/B\right)}\right)\right]. \end{array}$$

At large *B* this expression becomes the Maxwell–Jüttner distribution:$$N\left(b,\underline{k}\right)\underset{B\to \infty }{∼}C_{B}e^{-\underline{\beta }\cdot \underline{k}}.$$

Hence, going back to the original expression
$$\begin{array}{cc} N(b,\underline{k}) & =C_{B}{\left(\frac{b\cdot \xi }{B}\right)}^{-mcB}=C_{B}{\left(\frac{b^{0}}{B}\frac{k^{0}}{mc}-\frac{b}{B}\cdot \frac{k}{mc}+\frac{b^{4}}{B}\right)}^{-mcB}\\ \; & =C_{B}\;{\left(\frac{b^{4}}{B}\right)}^{-mcB}\;{\left(1+\frac{\underline{b}\cdot \underline{k}}{b^{4}mc}\right)}^{-mcB},\;\underline{b}:=\left(b^{0},b\right), \end{array}$$
and introducing
(59)$$q=1+\frac{1}{mcB}=1+\frac{\hbar \sqrt{\Lambda }}{mcn},$$

We finally obtain the Tsallis-type distribution
(60)$$N\left(b,\underline{k}\right)=C_{B}{\left(\frac{b^{4}}{B}\right)}^{-mcB}{\left(1-\left(1-q\right)\frac{B}{b^{4}}\;\underline{b}\cdot \underline{k}\right)}^{\frac{1}{1-q}}.$$

Analogously to (Equation 21) and all subsequent determinations of thermodynamical quantities, the following partition function is essential for their transcriptions to the de Sitter case: (61)$$\begin{array}{cc} Z_{\mathrm{dS}}\left(b,\underline{k}\right) & ={\left(\frac{b^{4}}{B}\right)}^{-mcB}\int_{V_{m}^{+}}{\left(1+\frac{\underline{b}\cdot \underline{k}}{b^{4}mc}\right)}^{-mcB}\;\frac{d^{3}k}{k_{0}} \end{array}$$(62)$$\begin{array}{cc} \; & =4\pi m^{2}c^{2}{\left(\frac{b^{4}}{B}\right)}^{-mcB}\int_{0}^{\infty }{\left(1+\left(\frac{b_{0}}{b^{4}}\right)\cosh t\right)}^{-mcB}\;\sinh^{2}t\;dt. \end{array}$$

With the following integral representation of the associated Legendre function of the First Kind $P_{\nu }^{\mu }\left(z\right)$ [26],
(63)$$P_{\nu }^{\mu }\left(z\right)=\frac{2^{-\nu }{\left(z^{2}-1\right)}^{-\mu /2}}{\Gamma (-\nu -\mu )\Gamma (\nu +1)}\int_{0}^{\infty }{\left(z+\cosh t\right)}^{-\nu -\mu -1}\;\sinh^{2\nu +1}t\;dt\;,$$
valid for z∉(−∞,−1] and Re(−μ)>Re(ν)>−1, the function (Equation 61) reads as
(64)$$Z_{\mathrm{dS}}\left(b,\underline{k}\right)={\left(8\pi \right)}^{3/2}\Gamma \left(1-mcB\right){\left(\frac{B}{b_{0}}\right)}^{mcB}{\left(\frac{B^{2}-b\cdot b}{b_{0}^{2}}\right)}^{mcB/2-3/4}\;P_{1/2}^{mcB-3/2}\left(\frac{b^{4}}{b_{0}}\right).$$

## 6. Conclusions

In this contribution, we have forged a groundbreaking link between the Tsallis distribution, quantum statistics, and the cosmological constant, illuminating the complex interplay between relativistic thermodynamics and a fundamental cosmological parameter.

Our key findings are encapsulated in Equations (Equation 59) and (Equation 60). The intricate technical details of the associated thermodynamic features (flux number, energy-momentum tensor, etc.) in the de Sitter space-time, along with their physical (and astrophysical!) implications and determinations (e.g., numerical factor(s) n), are reserved for future exploration. In this endeavor, analogous studies, such as those found in Refs. [27,28], may provide useful insights and avenues for the advancement of this project.

## Figures and Tables

**Figure 1 entropy-26-00273-f001:**
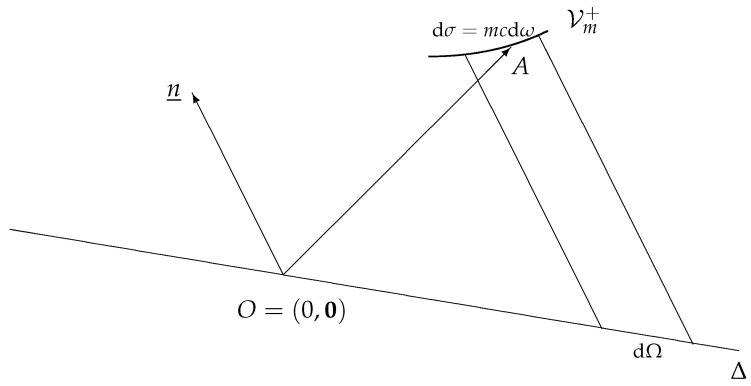
$\underline{n}$ is a time-like unit vector, Δ is a straight line passing through the origin and orthogonal (in the Minkowskian metric sense) to $\underline{n}$. The 4-momentum $\underline{k}=\left(k^{\mu }\right)=\left(k^{0},k\right)$ points toward a point *A* of the mass shell hyperboloid $V_{m}^{+}=\left\{\underline{k}\;,\;\underline{k}\cdot \underline{k}=m^{2}c^{2}\right\}$. dΩ is the length of the projection, along $\underline{n}$, of an infinitesimal hyperbolic interval at *A* of length dσ=mcdω.

**Figure 2 entropy-26-00273-f002:**
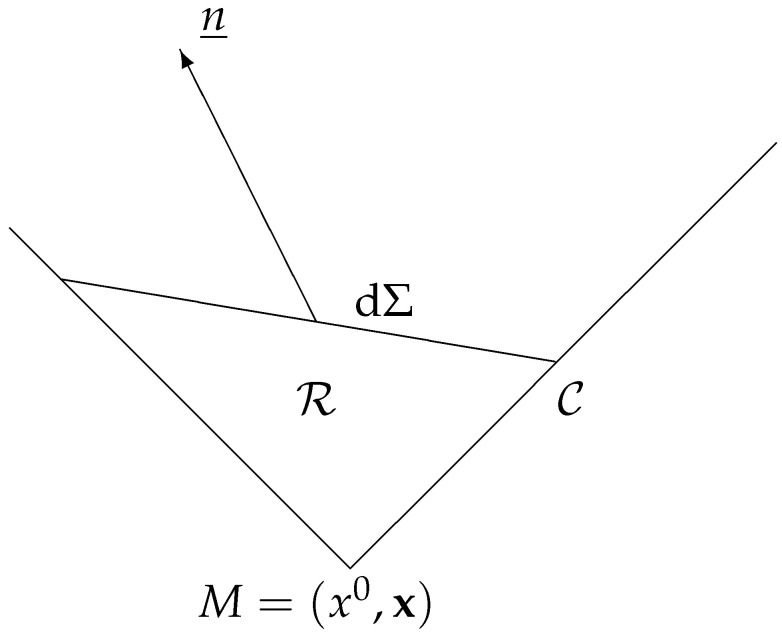
C is the portion of the null cone starting at the event $M=(x^{0},x)$ and limited by the infinitesimal space-like segment dΣ orthogonal to the time-like unit vector $\underline{n}$. R is the region delimited by *M*, the portion of the light cone C, and dΣ.

**Figure 3 entropy-26-00273-f003:**
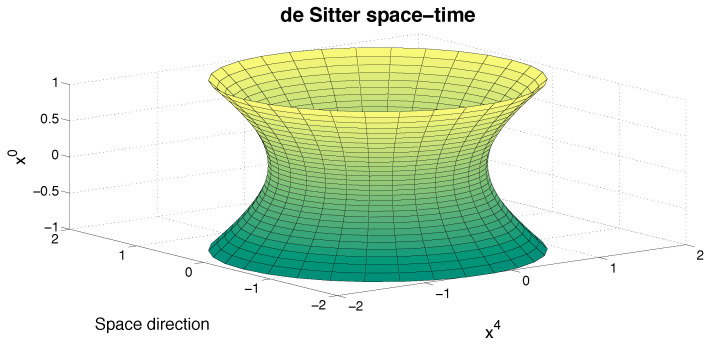
The de Sitter space-time as viewed as a one-sheet hyperboloid embedded in Minkowski space $M_{1,4}$.

## Data Availability

Data are contained within the article.

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
