# Peer review of "Tsallis Distribution as a Λ-Deformation of the Maxwell–Jüttner Distribution"

_entropy, 2024, doi:10.3390/e26030273_

Round 1

Reviewer 1 Report

Comments and Suggestions for Authors

Report of the Referee

Manuscript Ref.: Entropy-2916768 

Title: "Tsallis distribution as a Λ-deformation of the Maxwell-Jüttner

distribution"

=================================================

The author presents a very nice and interesting work on the obtention of Tsallis distribution from the relativistic Maxwell-Jüttner (MJ) distribution function. Starting from a relativisticaly covariant point of view for temperature, following de Broglie, the relativistic thermodynamics can be discussed in terms of an inverse temperature four-vector which is future-directed and timelike. It has been shown that the q-dependent Tsallis distribution can be considered as a de-Sitterian deformation of the MJ distribution. In this case, the parameter q is associated to the the cosmological constant Λ. The results presented in the manuscript open a new window for both physical and astrophysical implications. In addition, Tsallis-like distributions have been used extensively to describe initial conditions for particle production in relativistic heavy ion collisions.  Therefore, the demonstrations presented in the paper are of high interest in particle physics phenomenology at the Large Hadron Collider (LHC) and future hadron colliders.

The paper is very well presented and the references are quite adequate. The topic is hot and of interest for the researchers studying non-extensive distributions. The subject of the manuscript is a scientific breakthrough and the quality of the paper corresponds to the level of the journal “Entropy”. The manuscript is well-written, presenting ideas and methods clearly and analyzing the results thoroughly. The work is in suitable form for publication.

For the reasons presented above the paper is clear and well argued. It is suitable in my view for the Entropy journal.

----

I found some misprints and typos in the text (not exaustive):

Page 2, Line 43: Missing reference for Synge.

Page 3: Missing ponctuation in Eq. (12).

Page 6, Line 139: "space-tiem"--> space-time.

Page 6: Double (extra) comma in Eq. (36).

Comments on the Quality of English Language

Minor editing of English language required.

Author Response

I am appreciative of the Reviewer for the favorable evaluation of the content of my submitted article.  In response to Reviewer's recommendations, I have improved  the abstract and rectified any identified misprints, as is shown in the red-highlighted version of the revised manuscript.

Reviewer 2 Report

Comments and Suggestions for Authors

It is a nice piece of mathematical work, with sound result. It is unfortunate that a numerical factor still remains in the correspondence between q and other physical quantities, cf. the cosmological constant.

Some references, later than the already cited ones, may actualize the contribution further. About relativistic thermodynamics I call the attention to EPL 89, 30001, 2010, and about q related to de-Sitter lambda to  PLB 726, 861, 2013; and PLB 708, 276, 2012.

Author Response

I sincerely thank the Reviewer for their positive evaluation of the content of my submitted article. In light of their recommendations, I have briefly mentioned the question concerning the numerical factor, and I have duly included the relevant references as suggested, as is shown in the red-highlighted version of the revised manuscript.

Reviewer 3 Report

Comments and Suggestions for Authors

This is an insightful study that establishes a connection between Tsallis Statistics and the Cosmological Constant, drawing on considerations regarding the behavior of thermodynamic quantities under Lorentz Transformation and the effects of the de Sitter space curvature.

I strongly recommend accepting and publishing the paper as soon as possible. Please note the missing reference number in line 43.

Author Response

I am grateful to the Reviewer for the positive assessment on the content of the submitted article. I have followed Reviewer’s recommendations concerning the missing reference.